# Semantic 3D Mapping from Deep Image Segmentation



**Francisco Martín** [1,*] **, Fernando González** [1] **, José Miguel Guerrero** [1] **, Manuel Fernández** [1] **and Jonatan Ginés** [2]

1   Intelligent Robotics Lab, Rey Juan Carlos University, 28943 Fuenlabrada, Spain;
    f.gonzalezr.2016@alumnos.urjc.es (F.G.); josemiguel.guerrero@urjc.es (J.M.G.);
    manuel.fernandezc@urjc.es (M.F.)
2   Escuela Internacional de Doctorado, Rey Juan Carlos University, 28933 Móstoles, Spain; jonatan.gines@urjc.es
*   Correspondence: francisco.rico@urjc.es

**Abstract:** The perception and identification of visual stimuli from the environment is a fundamental capacity of autonomous mobile robots. Current deep learning techniques make it possible to identify and segment objects of interest in an image. This paper presents a novel algorithm to segment the object's space from a deep segmentation of an image taken by a 3D camera. The proposed approach solves the boundary pixel problem that appears when a direct mapping from segmented pixels to their correspondence in the point cloud is used. We validate our approach by comparing baseline approaches using real images taken by a 3D camera, showing that our method outperforms their results in terms of accuracy and reliability. As an application of the proposed algorithm, we present a semantic mapping approach for a mobile robot's indoor environments.

**Keywords:** image segmentation; deep learning; 3D semantic mapping





## 1. Introduction

There is a great interest in creating intelligent behaviors in social robots. Social robots interact with humans or objects to carry out tasks that usually involve moving around their environment. In these tasks, vision is essential to detect elements of the environment to interact with. A few years ago, image processing modules were ad-hoc developments, lacking from generalization. In recent years, deep learning techniques based on convolutional neural networks have revolutionized how a robot extracts information from images. Initial approaches allowed to calculate bounding boxes on images, labeling detected elements with specific class labels. New advances in hardware allowed to enormously speed up training and networks detection times using GPUs. In addition, extensive datasets allowed to train networks with a vast amount of real data, increasing the detections' generality and robustness. Current approaches further enable labeling specific pixels inside the image's bounding boxes with its detected class label. This is known as deep image segmentation. Today, these approaches' robustness can be tested in various available implementations, such as Yolact [1] or MSeg [2].

In Robotics, it is essential to perceive the world as it is, in three dimensions. In Artificial Vision, the techniques for 3D reconstruction of a scene have required stereoscopic cameras or movement analysis techniques for years. The appearance of RGB-D cameras a few years ago has been a revolution in many applications, especially in robotics. These sensors have similar size and cost to a standard camera, but also provide 3D position of each pixel in the captured image. Typically these sensors produce point clouds, where each point is on a 3D coordinate and contains color information. A robot can detect shapes in the environment and relate them to its position using these sensors. A robot can perceive planes that correspond to the floor or walls, detect if a door is open, the distance to a person, or the presence of small obstacles in robot's path that cannot be detected with a 2D laser sensor unless it is not right at the right height.

RGB-D sensors have the advantage that they link each 3D point with its corresponding 2D coordinate in the image. Taking advantage of this feature, we have developed a

point cloud segmentation technique based on the segmentation of its corresponding 2D image using deep learning techniques. The novelty of our approach is how we deal with the boundary pixel problem. This problem arises when obtaining the 3D position of segmentation's border pixels. Small miscalculations in the mapping or rapid movements may map these border pixels to the scene's background rather than to the segmented object 3D coordinates. This problem produces outliers that negatively impact when calculating and recording the space that the object occupies.

Another novelty of our approach is that the output of our algorithm uses octomaps to represent the segmentation. An octomap efficiently represents probabilistic 3D space occupancy using octomaps [3]. An octomap is a hierarchical data structure for spatial subdivision in 3D. Each node in this tree, called voxel, represents a portion of space. Each octomap leaf is a voxel that contains the occupancy probability of the space it represents. It is much more efficient than a point cloud in terms of access, storage, and information transmission [4]. We use an octomap variant that includes color information in the voxels, key to detecting its object class. The advantage of this decision is that it can be easily debugged by visual tools. A single octomap can contain all detected objects, distinguishing each object's class.

In this paper, we show how to use our approach to semantically map a robot's environment with segmentation information. The semantic mapping problem can be defined as the process of building a map which associates spatial concepts (i.e., labels) with spatial entities (such as objects) [5]. There are several potential applications of semantic mapping in robotics. For example, a robot could use this information to self-locate in its environment, improving accuracy or resetting its position in the case of local (tracking) location algorithms. Or it could also be used to command the robot to move close to a previously detected object instead of a specific metric position.

In summary, this paper has two main contributions:

- Two novel algorithms to calculate the space occupied by elements detected from deep image segmentation.
- Their application to the semantic mapping problem, which is used to validate its performance.

The rest of the paper is organized as follows: In Section 2, we introduce relevant works related to our approach. Then, in Section 3, we describe our work in 3D segmentation, which is later used in Section 3.2 to map the environment with semantic information. We validate our approach in Section 4, and finally, after presenting some limitation in Section 5, we provide the conclusions in Section 6.

## 2. Related Work

### 2.1. Semantic Mapping

RBG-D cameras are the main tool for semantic mapping. These provide affordable volumetric segmentation supported by techniques usually focused on conventional images. Based on that initial segmentation process, authors like Grinbald [6] perform additional tasks such as including scene elements and predictions over the frames, to finally integrate information about 3D shapes, location, and, if available, semantic classes into a global volume.

Semantic mapping is a central component for robot localization, so it is usually integrated with SLAM implementations, such as in [7]. Predictions from a Convolutional Neural Network can be combined into a dense SLAM map, like in [8], and even later include geometric information from depth images into the scene processing [9]. Other works like [10] work at object-level, building dense semantic maps based on preexisting 3D RBG-D-based SLAM frameworks like [11].

Other works rely on an object database to create object-oriented maps recognizing those objects from RGB-D sources and supporting the SLAM process. The database can be built during the exploration, like in [12], based on geometric segmentation information acquired from the cameras. In general, semantic SLAM builds its own semantic objects

based on the entities created in the mapping phase [13]. These semantic 3D mapping approaches face serious complexity issues when it comes to large, dynamic environments. Berrio et al. [14] proposed using octomaps to leverage the overall efficiency. In addition, Yang in [15] used scrolling grids in dynamic scenarios.

### 2.2. Object Detection and Segmentation

Several authors base their work on single-stage detectors like YOLO v2 [16], YOLO v3 [17], or ResNet [18] and add depth information to the network. For example, Harizas et al. in [19] based their work on SegNet [20]. They duplicate the encoder part, extracting features both from the RGB and depth images, to finally fuse both as the network decodes them. Ophoff in [21] compared how fussing depth and RGB features at different layers resulted in better performances for a YOLO v2 CNN. However, their output is still a bounding box, not including any 3D information in the end. Linder et al. in [22] modified a YOLO v3 CNN to include depth information and produce a 3D centroid of human detections. Others, like [23] proposed an architecture that deals with both 2D and 3D object detection, supported by a ResNet-18 CNN.

Alternatively, Wang et al. in [24] introduced depth-aware convolution and depth-aware average pooling operations for a new kind of CNN. These operations avoid the introduction of additional parameters, and keep the computation complexity to the conventional CNN.

Song et al. in [25] addressed the problem using a 3D CNN. Initially limited in scalability, other recent works are improving the initial memory and computational costs derived from a pure 3D approach. Charles et al. [26] created PointNet, a 3D k-nearest neighbor (kNN) graph neural network based on point cloud data which achieved state-of-the-art performance on RGB-D segmentation. Real-time instance segmentation has been finally achieved by Yolact [1], a method based on the use of RetinaNet using ResNet-101 and Feature pyramid networks as default feature backbone. In addition, MSeg [2] offers robust real time segmentation, plus a single, composite dataset combining the largest semantic segmentation datasets.

These deep learning approaches have been interfaced with ROS [27] to provide semantic segmentation in robotic applications, e.g., wrappers for Deepnet [28] or custom CNNs based on PyTorch [28] or Tensorflow [29]. Their main drawback is that the output is always given as images, limiting their use in 3D applications.

## 3. Segmentation and Mapping in 3D

As described in the previous section, multiple works on deep 2D image segmentation have their software available as libraries, executable programs, or as ROS nodes, ready to be integrated into any application. These implementations are usually already pre-trained with general-purpose datasets. In this work, Yolact [1] will be used, although any segmentation software can used as long as its output labels in a complete image or separate sub-images, matching each 2D pixel with a class. Figure 1 shows examples of 3D segmentations performed on the image.

The general architecture of our approach can be seen in Figure 2. RGB-D camera provides an image and its corresponding point cloud. An image $I$ is a matrix of $W \times H$ pixels, each containing the three color components $< r, g, b >$. The point cloud $P$ is a vector of length $l$ that contains the tuple $< x, y, z, r, g, b >$ with the spatial coordinates $(x, y, z)$, and the color components. If the point cloud is registered, the point cloud $l_P$ has a size of $W \times H$ tuples. Here, pixel position in the image $p_I(i, j)$ corresponds to the point $po_P(k), k = i * W + j$. Those points $po_P$ where it has not been possible to calculate the spatial information will establish their three spatial coordinates to infinity.

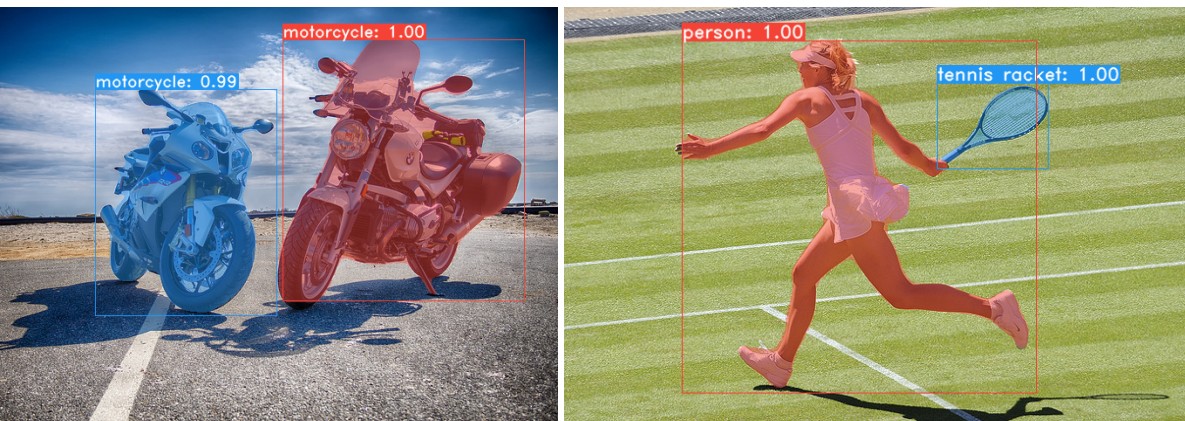

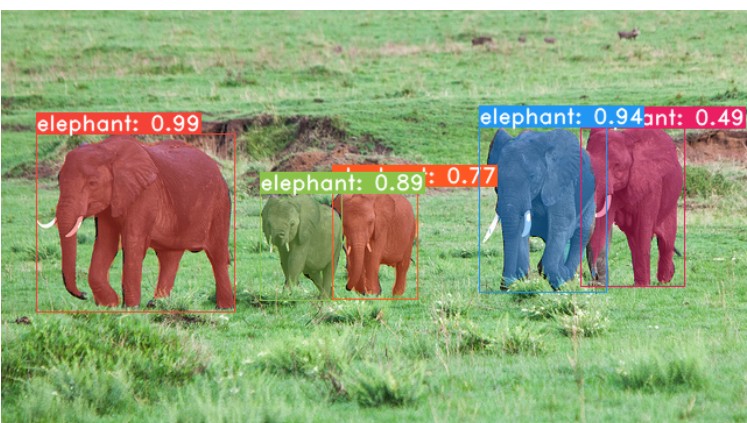

**Figure 1.** Example of deep image segmentation.

This set of bounding boxes is the input to our 3D segmentation system. This system uses the point cloud corresponding to the segmented image to obtain the 3D position of pixels that belong to the segmentation. The result includes all the detections in the image, and represents them in three different ways: as a point cloud, as a 3D bounding box, and as an octomap. As there is no standard format for 3D segmentation information, offering multiple output formats allow us to cover the most common ones used in other applications, while reducing potential delays due to cascade conversions.

Color is used as the key to determining which class the detections belong to in the processing. If the chosen representation format is octomap, for example, all voxels representing space occupied by a class's elements will have the same color.

The semantic mapping system uses octomap representation of detections as input. As explained before, an octomap can include probabilistic information, useful in the mapping process. The result of the mapping will be an octomap containing the probabilities of object classes in different positions in space on the map's reference axis. This map can be stored and loaded for further use by other applications.

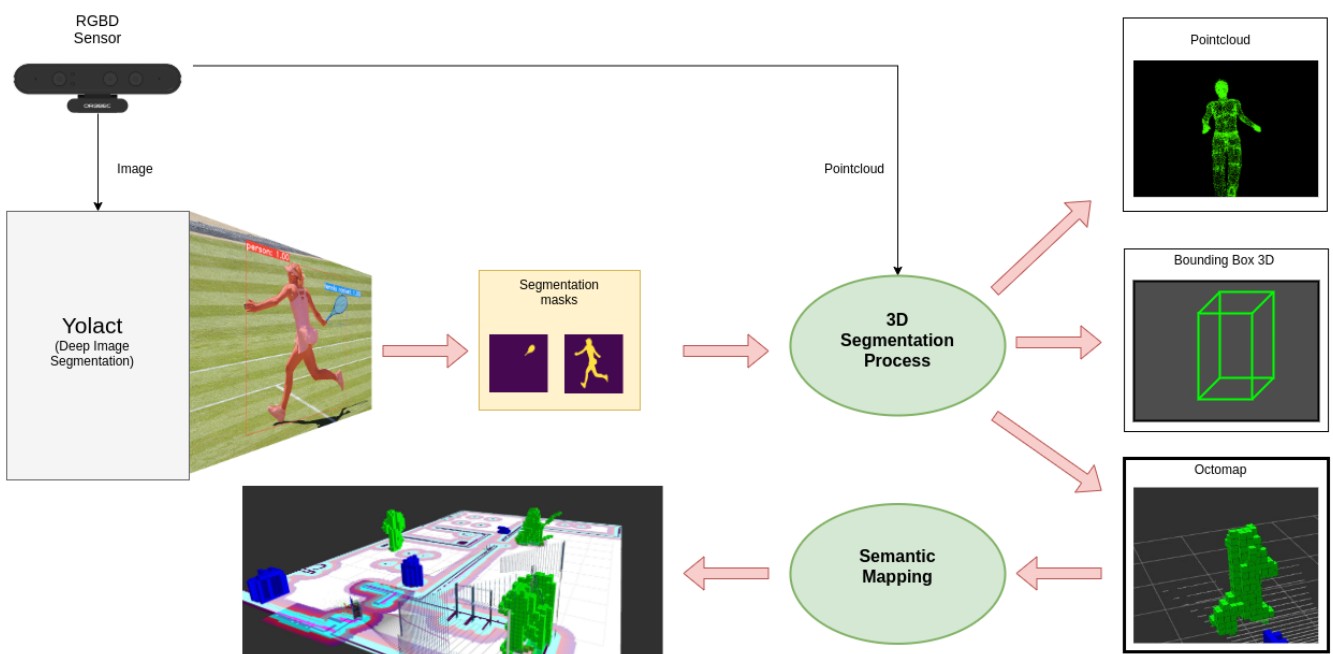

**Figure 2.** Complete process of our approach.

### 3.1. 3D Segmentation

3D segmentation from a segmented image and its corresponding point cloud is not simple. If we simply accumulate the points in the point cloud corresponding to each pixel in a mask, we will obtain a degraded result due to the *boundary point* (Figure 3) problem. This problem arises when the depth image does not exactly match the color image where segmentation is performed, mainly due to errors or lack of accurate calibration. This problem is critical because it may place targets far from where they are.

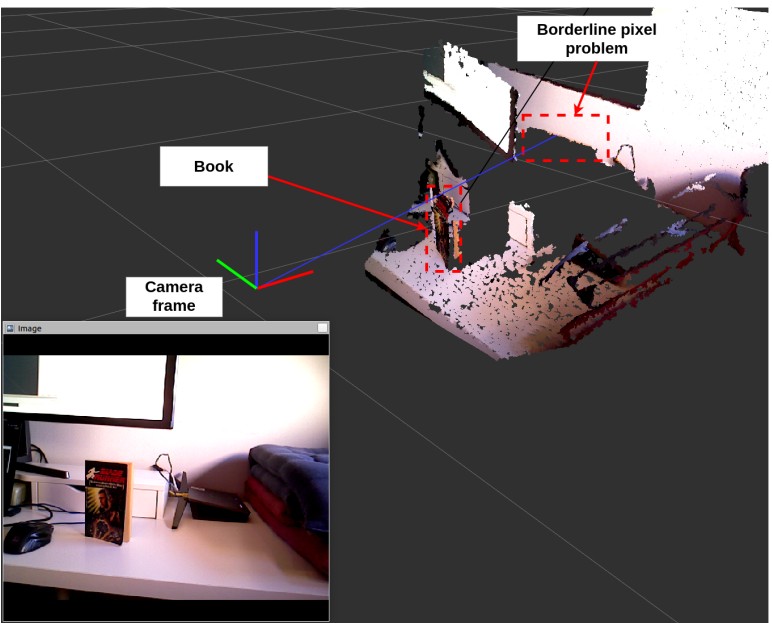

**Figure 3.** At the bottom left, we show the segmented image in RViz https://github.com/ros-visualization/rviz (accessed on 21 January 2021). The rest of the figure shows the corresponding point cloud. The *Boundary point* problem appears at the edges of the segmented areas. As seen in the point cloud, some points belong to the book whose positions are on the back wall.

The algorithm calculates an octomap with all the detections from a segmented image and its corresponding point cloud. Each detection in the octomap has a color that determines what kind of object it belongs to. Using color as the key to indicate the detected object has several advantages:

- The color of each object class is determined by 2D segmentation software, and it is not repeated.
- The octomap allows assigning a color to each voxel of the octomap, so that the detections are uniquely marked.
- It allows to visualize the octomap in a simple and intuitive way.

Our 3D segmentation algorithm, presented in Algorithm 1, works as it follows:

1. **[line 2]**: The function *deep_2d_segment* returns a set of detections in the image. Each detection is a 2D bounding box containing a mask. Each pixel is the mask corresponding to the detected object, marked with the object class color as shown in Figure 2.
2. **[line 3]**: The result of the algorithm will be stored in *detection_octomap*, a colored octomap with each voxel containing a color and a probability.
3. **[lines 4–7]**: This loop iterates over each one of the detections in the image:

    (a) **[line 5]**: This algorithm uses a KD-tree [30] that uses a fast neighbor search algorithm [31]. This data structure is initialized with a point cloud, and allows querying which points are in the vicinity of a reference point. By having the point cloud registered with the image (the $i$th pixel of the image corresponds to the $i$th point of the point cloud) and the pixels that belong to a detection, we can create point clouds that contain only the 3D points of the detection made in the image.

    (b) **[line 6]**: Processing begins with a 3D point that belongs to the detected object. This 3D point is the result of obtaining the position of the point cloud that corresponds to one of the detection pixels in the image. The detection center or centroid is valid for this purpose.

    (c) **[line 7]**: Start the recursive processing to ensure octomap connectivity.

The recursive function *EXPAND_octomap*, also presented in Algorithm 1, is called until all points representing the detected object in the output octomap are connected. As arguments, it receives the 3D point to expand, the KD-tree with the detected points and the octomap to store the result. It works as it follows:

1. **[lines 2,3]**: Neighbors of the input point are calculated using a search in a radius $kr$. $kr$ is the voxel size in the output octomap.
2. **[lines 4,5]**: If a neighbor is found, it is included into the output octomap. Note that the output octomap will contain points that were not in the original detection point cloud, but rather represent a continuous space occupied by this point cloud.
3. **[lines 6–9]**: This function will be called recursively with the expansion points that are around the input point, to expand the output octomap in all axes.
4. **[lines 10,11]**: If the expansion point is already in the octomap output recursion to this point is stopped. If not, the recursive function is called.

Figure 4 shows the application of this algorithm in a point cloud. For simplicity and better visualization, we have removed one dimension, so we will see a 2D projection of Algorithm 1. Let us imagine that this is a detection seen from above (Figure 4a), in which the detected object corresponds to the blue points. Red dots are errors in detection:

1. In Figure 4b, the starting point is established and the neighbor search is performed.
2. In Figure 4c, when finding neighbors (orange points from the original point cloud), their position is added to the output octomap (green box), new search points (green points) are created and the process is repeated in each of them.
3. Any successful searching produces a new cell in the output octomap, and generate new search points (Figure 4d).

4. The process is repeated in Figure 4e–g, until the detection limits are reached, or until there are no new search points that can be generated (green points with no candidates within the search radius).

5. The final result (Figure 4h) is an octomap that includes the points that really belong to the object to be detected. Outliers are not included in this octomap.

---

**Algorithm 1** 3D Segmentation

---

1: **function** 3D_Segmentation(*input_image, input_cloud*)
2:      *2d_detections* ← *deep_2d_segment*(*input_image*)
3:      *detection_octomap* ← ∅
4:      **for all** *detection* ∈ *2d_detections* **do**
5:          *KD − tree* ← *get_KD − tree*(*input_cloud, detection*)
6:          *start_point* = *to_3D*(*detection.center.x, detection.center.y*)
7:          *expand_octomap*(*start_point, KD − tree, octomap*)

1: **function** EXPAND_OCTOMAP(point, KD-tree, octomap)
2:      *kr* ← *KD − tree.resolution*
3:      *neighbours* ← *KD − tree.radious_search*(*point, kr/2*)
4:      **if** *neighbours* ≠ ∅ **then**
5:          *octomap* ← {*point, KD − tree.color*}
6:          **for** *dx* ← {−*kr, kr*} **do**
7:             **for** *dy* ← {−*kr, kr*} **do**
8:                **for** *dz* ← {−*kr, kr*} **do**
9:                    *new_point* = {*point.x + dx, point.y + dy, point.z + dz*}
10:                  **if** *new_point* ∉ *octomap* **then**
11:                    *expand_octomap*(*new_point, KD-tree, octomap*)

---

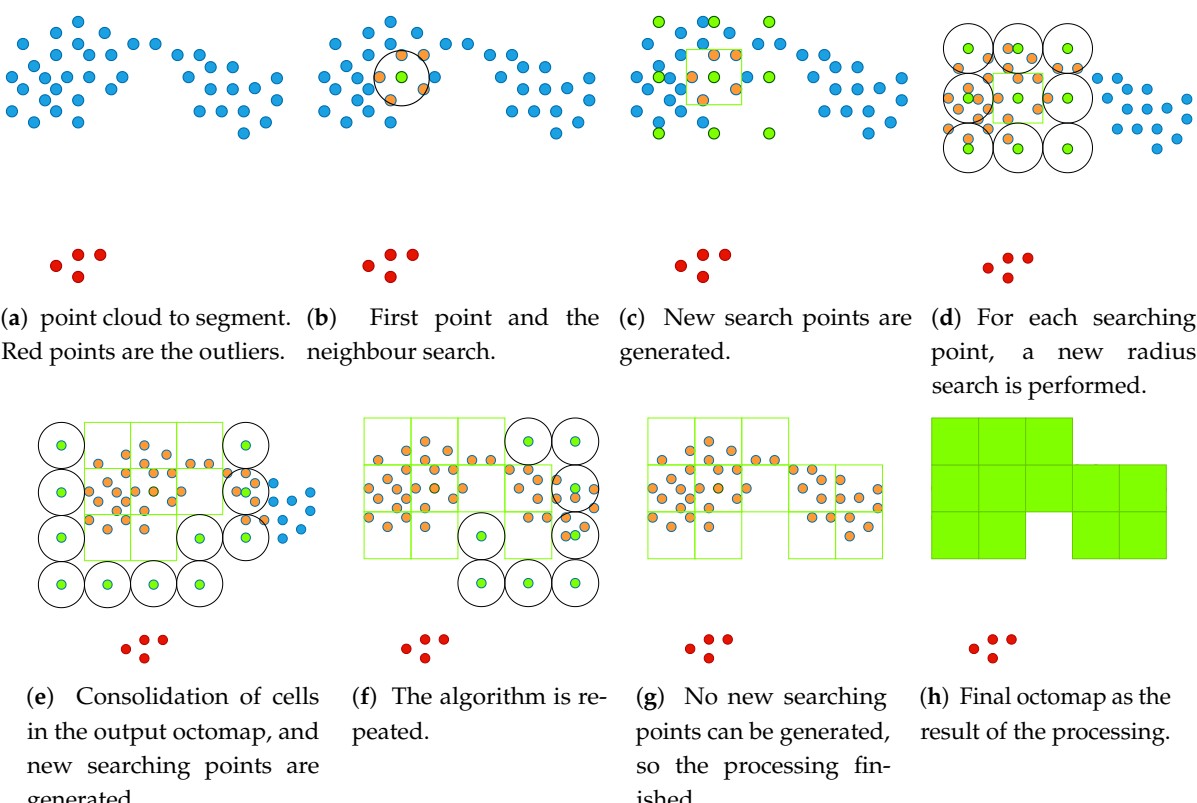

(**a**) point cloud to segment. Red points are the outliers.

(**b**) First point and the neighbour search.

(**c**) New search points are generated.

(**d**) For each searching point, a new radius search is performed.

(**e**) Consolidation of cells in the output octomap, and new searching points are generated.

(**f**) The algorithm is repeated.

(**g**) No new searching points can be generated, so the processing finished.

(**h**) Final octomap as the result of the processing.

**Figure 4.** Example of the application of the algorithm with a simplification in 2D.

### 3.2. Semantic Mapping

The semantic mapping adds spatial occupation information of the detected elements to the environment map using the previous section's algorithm. The starting map is a 2D occupation grid where each cell represents the probability of an obstacle. Figure 5 shows where the obstacles are in the dark areas, the clear region is white, and the gray area is unknown.

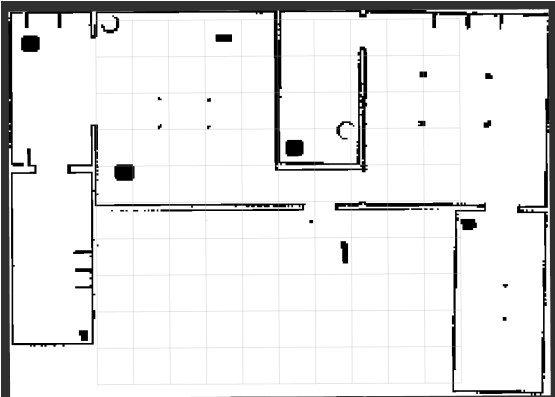

**Figure 5.** Grid map of the simulated environment.

Each axis of reference is named, and we will call it a frame, e.g., the frame where the map is located is the *frame map*. More frames are related to each other by an invertible transformation matrix $RT_{4\times4}$ representing a translation and a rotation. The frames form a tree of frames, where each node is a frame, and the arcs are geometric transformations of translation and rotation. There can be no loops in this tree, and each frame has only one predecessor in the tree. In a robot, for example, there is one for each sensor or joint. The root of a robot's structure is called by convention *base_footprint* (Figure 6), and it is at ground level, in the center of the robot. Any 3D coordinate in any frame can be transformed to coordinates in another frame. For example, the coordinate of an obstacle perceived in the *laser frame* can be transformed into a coordinate of the *base_footprint frame*. The root of a robot's frame tree is the *odom frame*, whose child is *base_footprint frame*. The arc that joins them indicates the perceived displacement since the robot was started. This odometry is updated based on the robot's proprioceptive sensors.

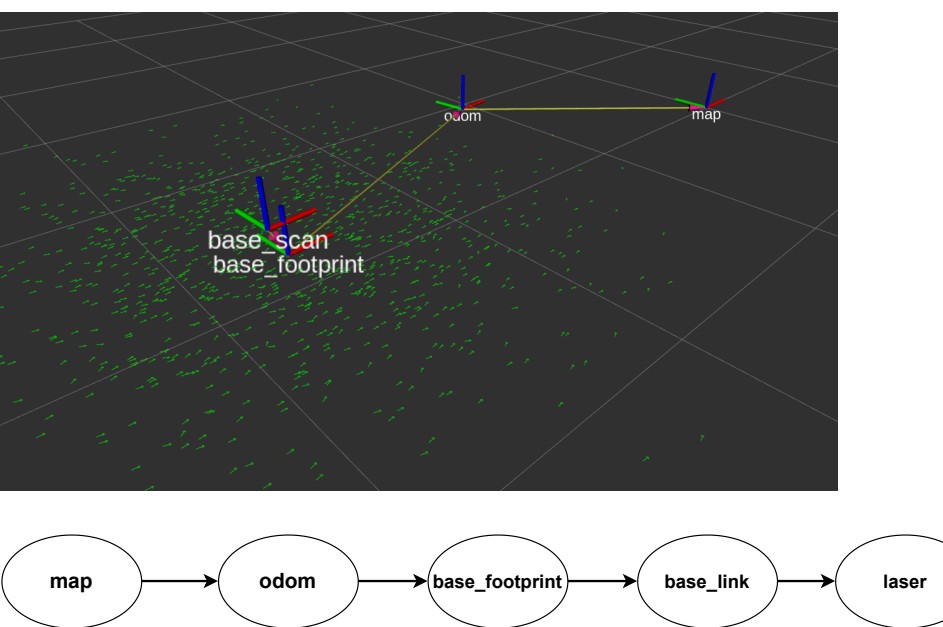

**Figure 6.** Coordinate axes connections from the map frame to the sensor frame.

In this work, the robot is located using a particle filter. The results of this localization process are:

- The connection of the frame map with the root of the robot tree, the frame odom: $RT^{map \to odom}$.

- A covariance matrix $E_{6 \times 6}$ that indicates the precision of the robot's location ($x, y, z, roll, pitch, yaw$) in space.

$\mathcal{O}_t$ is the octomap that contains the cells that correspond to the objects detected in the 3D segmentation process described in the previous section. Each cell is a voxel in the space with an object class (coded by its color) and a probability. In the semantic mapping, we maintain a semantic map as an octomap $\mathcal{M}_t$ in *frame map*, that updates with each instant perception $\mathcal{O}_t$ (Figure 7).

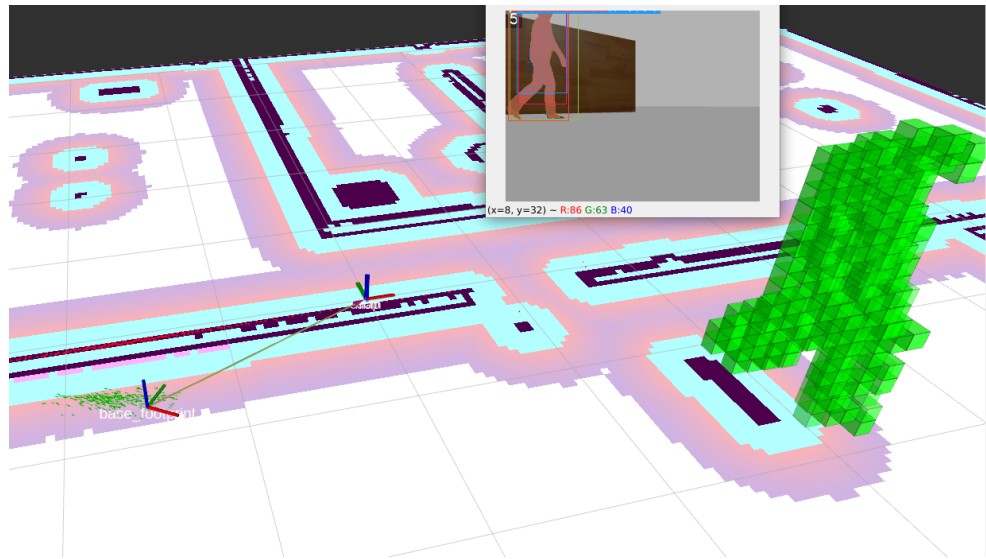

**Figure 7.** Instantaneous perception $\mathcal{O}_t$ from the 3D segmentation process.

Perceptions $\mathcal{O}_t$ are located in the frame of the sensor with which they were captured. If a robot is located, it is possible to obtain each cell of the octomap in the frame map, since there is a path in the frame tree from the sensor frame to the frame map, as shown in Equation (1).

$$RT^{map \to sensor} = RT^{map \to odom \to base\_footprint \to base\_link \to sensor} =$$

$$RT^{map \to odom} * RT^{odom \to base\_footprint} * RT^{base\_footprint \to base\_link} * RT^{base\_link \to sensor}$$

$$p^{map} = (RT^{map \to sensor})^{-1} * p^{sensor}, \forall p \in \mathcal{O}_t \tag{1}$$

We can not register $p^{map}$ directly in $\mathcal{M}_t$ because:

- There is an uncertainty in $RT^{map \to sensor}$ since the particle filter estimates the robot's position with uncertainty represented by the covariance matrix **E** of the location.
- There may be erroneous perceptions sporadically due to some error in the detection of the objects in the image, or in some synchronization between the point cloud and the image.

For this reason, we have developed an algorithm that incrementally composes new octomaps taking into account the uncertainty in the location of the robot. The application steps for each new $\mathcal{O}_t$ are as follows:

1.  Obtain the current uncertainty E of the location.

$$\mathbf{E} = \begin{pmatrix} \sigma_{xx}^2 & \sigma_{xy}^2 & \cdots & \sigma_{xw}^2 \\ \sigma_{yx}^2 & \sigma_{yy}^2 & \cdots & \sigma_{yw}^2 \\ \vdots & \vdots & \ddots & \vdots \\ \sigma_{wx}^2 & \sigma_{wy}^2 & \cdots & \sigma_{ww}^2 \end{pmatrix} \tag{2}$$

2.  Generate a random noise transform $RT'$ from the uncertainty using normal distributions of probability

$$Noise_{Translation} = random(\mathcal{N}(0, \sigma_{xx}^2), \mathcal{N}(0, \sigma_{yy}^2), \mathcal{N}(0, \sigma_{zz}^2))$$
$$Noise_{Rotation} = random(\mathcal{N}(0, \sigma_{rr}^2), \mathcal{N}(0, \sigma_{pp}^2), \mathcal{N}(0, \sigma_{ww}^2))$$
$$RT' = (Noise_{Translation}, Noise_{Rotation}) \tag{3}$$

3.  Calculate a new noisy transform $RT'^{map \rightarrow sensor}$

$$RT'^{map \rightarrow sensor} =$$
$$RT^{map \rightarrow odom} * RT' * RT^{odom \rightarrow base\_footprint} * RT^{base\_footprint \rightarrow base\_link} * RT^{base\_link \rightarrow sensor} \tag{4}$$

4.  We repeat N times the steps 2–3 for each $\mathcal{O}_t$. We update $\mathcal{M}_t$ from $\mathcal{M}_{t-1}$ accumulating the probabilities of each cell as follows, being $K$ a constant in range [0–1]. If $K$ is near 1, the previous value in the cell has more importance than the new perception when fusing both values:

$$\mathcal{M}_t = \sum_{i=1}^{N} (\frac{K}{N} * \mathcal{M}_{t-1} + \frac{1-K}{N} * RT_i'^{map \rightarrow sensor} * \mathcal{O}_t) \tag{5}$$

The result is shown in Figure 11d. $\mathcal{M}_t$ is a octomap in which each cell contains the class of an object coded as a color, and the occupancy probability.

## 4. Results

In this section, we present the work we have done to validate our contributions. In the first experiment, we will quantitatively compare our proposal with other common approaches, obtaining precision and reliability measures. In the second experiment, we will show the semantic mapping application results with a mobile robot.

### 4.1. Experiments of the Algorithm for 3D Segmentation

The first experiment measures the precision with which an octomap represents a segmented object in the image. The proposed algorithm correctly detects the voxels of the octomap that belong to the detected object, and what we are going to show is how other approximations deteriorate this result. The first of these approximations corresponds to the original algorithm, in which a direct translation of pixels and voxels is made (suffering the problem of *borderline pixels*). The second approximation is when erosion is carried out to the segmentation in the image, trying to eliminate the border pixels, before translating to voxels.

#### 4.1.1. Methodology

For this experiment, we have recorded all the information of a sequence of a person in front of the camera. We record this information in a file called bag file, using the rosbag tool. This file contains all the sensor readings and the geometric relationships between frames labeled with the timestamp when they occurred. This information includes the frames, 2D image, point cloud, and the segmentation process's output. For each algorithm to compare, we have reproduced this bag file to compare the results with the same inputs.



### 4.1.2. Results

Figure 8 shows the amount of voxels obtained in 715 images extracted from the video. As we can observe, in our method, the octomaps' number detected is slightly lower than the original approach. That means that our approach removes the octomap outliers appearing in the 3D detection while keeping the real octomaps belonging to the detected person. The erosion method removes too many octomaps from the detected person, underperforming the proposed method.

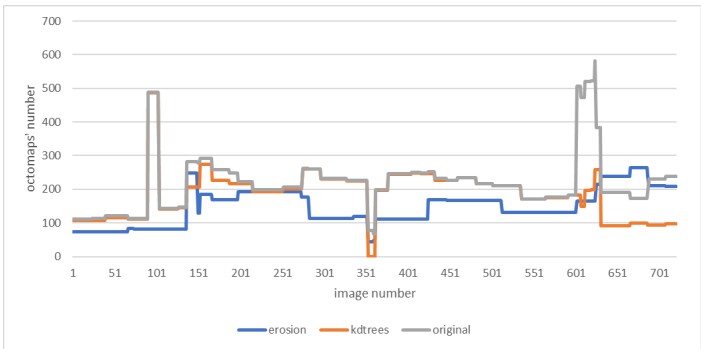

**Figure 8.** Comparison of different methods.

Differences in the amount of detected octomaps on each image are shown in Figure 9. This figure shows how the original method compared with the proposed method has a similar number of octomaps detected of the person, but a few more due to the outliers we want to remove. On the other side, the erosion method obtains less than necessary octomaps due to the image processing used to reduce the person's skeleton.

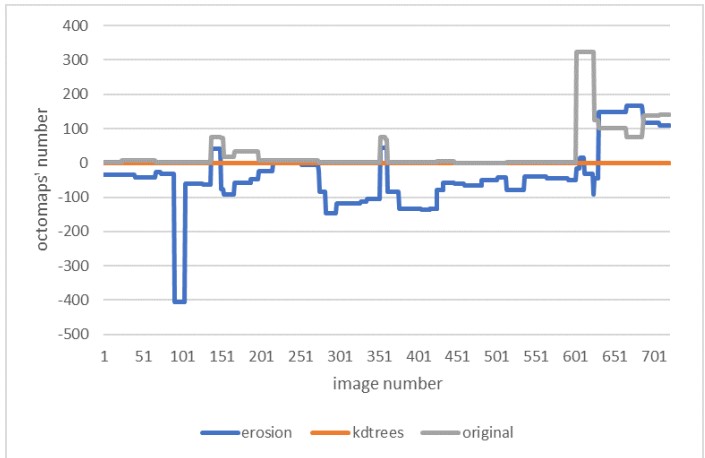

**Figure 9.** Differences of erosion and original with the proposed method.

Figure 10 shows processing times along the video in seconds. As we can observe, in the erosion method (Figure 10a), this time is higher than in our approach (Figure 10b). In addition, the erosion method has completed 61 iterations, while our method has completed 113 iterations due to shorter processing times.

Some statistics about the processing time are shown in Table 1. As expected, the proposed method has lower average processing times and variation, consistently overperforming the erosion method.

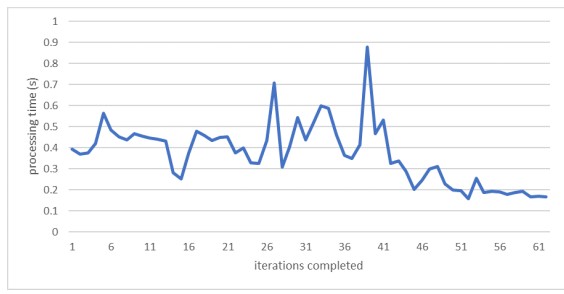

(**a**) Seconds in each cycle for the erosion method.

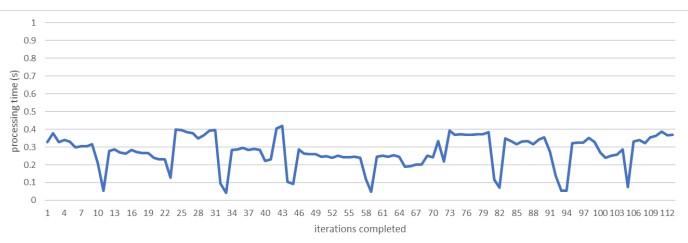

(**b**) Seconds in each cycle for the proposed method.

**Figure 10.** Processing times in seconds.

**Table 1.** Statistics of processing times for erosion and the proposed methods in seconds.

| Stat. | Erosion | Proposed Method |
|:---:|:---:|:---:|
| Maximum | 0.88 s | 0.42 s |
| Average | 0.37 s | 0.27 s |
| Minimum | 0.15 s | 0.04 s |

### 4.2. Application: Mapping a Domestic Environment

In the second experiment, we semantically mapped a real home environment. Items of interest are shown in Table 2. This experiment illustrates how the mapping process can be integrated along with other robot capabilities in a real scenario.

**Table 2.** Classes of objects used in the mapping process, with their associate colors.

| Class | Color |
|:---:|:---:|
| Cup | FF0000 |
| Bottle | 00FF00 |
| Chair | 0000FF |
| Oven | FF00FF |
| Tv | 00FFFF |

Figure 11 shows the mapping process of a segmented image during the second experiment. We have used a wheeled robot (Figure 11a) equipped with a 360° laser RPLidar A-2 and an Orbbec Astra RGB-D camera. The computer is an Intel Core i7 2.2 GHz with a GPU NVidia GeForce GTX 1070.

Detections are transformed to the frame map and inserted in the semantic map as octomap cells, each with a color representing its class and a probability, as shown in Figure 11b,c. The final map is shown in Figure 11d.

Our method matches other state-of-the-art 3D segmentation approaches while it does require complex operations. For example, Wang et al. [24] achieved accuracies around 70% using depth-aware convolutional operations, thus intensive training. Song et al. [25] also obtain precisions around 75% training models with occupancy and multi-class labeling, but it requires training from scratch with a large-scale dataset of synthetic 3D scenes with dense volumetric annotations.

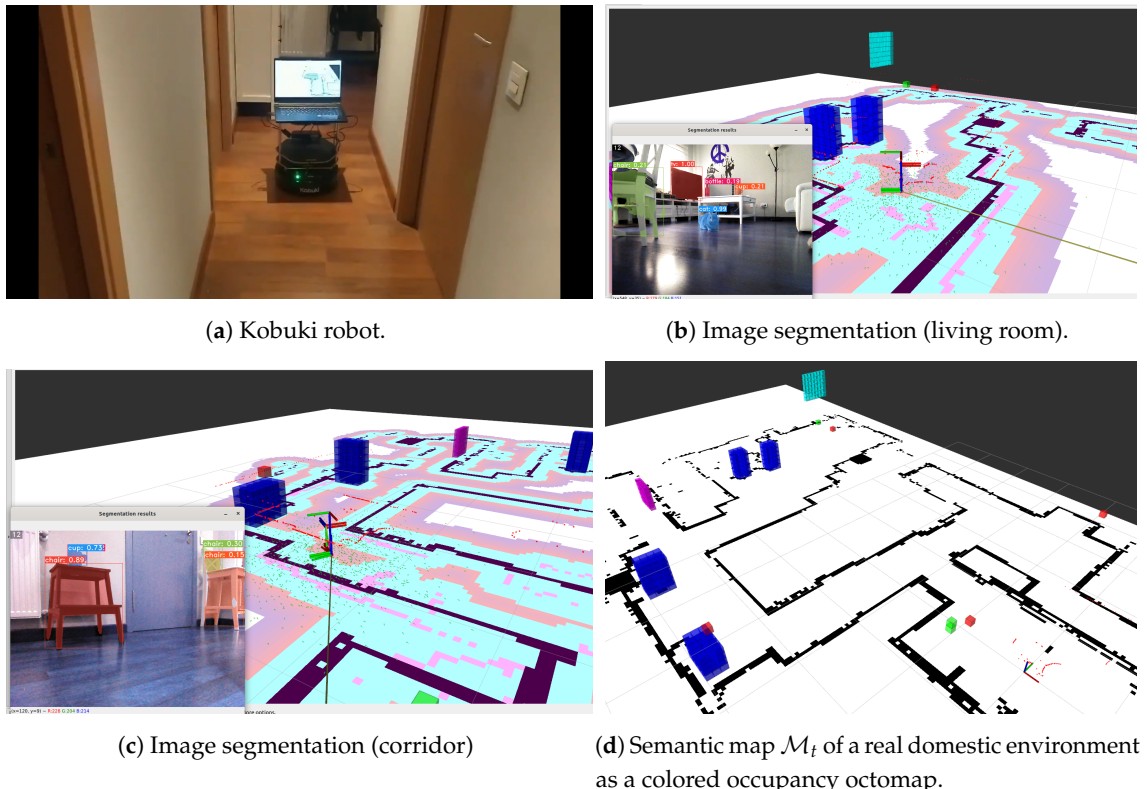

(**a**) Kobuki robot.

(**b**) Image segmentation (living room).

(**c**) Image segmentation (corridor)

(**d**) Semantic map $\mathcal{M}_t$ of a real domestic environment as a colored occupancy octomap.

**Figure 11.** Mapping of a real domestic environment.

## 5. Limitations

The 3D segmentation algorithm proposed in this work is efficient. It considerably improves the results obtained with a direct correspondence, or a preprocessing of the image, as we have shown in the previous section's experiments. However, it has certain limitations. These limitations arise from the fact that the method is based on the connection of 3D points corresponding to a segmented object in an image. We assume that the points that belong to the object have a certain continuity or neighborhood in space and that the points that do not really belong to the object are far from the rest of the detection points. These assumptions solve the *boundary point* problem in virtually all cases, but false positives or false negatives could appear in the final result if these assumptions are not met. Another limitation of our approach is related to the point in space at which the expansion algorithm starts. If this point does not belong to the object to be detected, the algorithm will fail. This error can happen if the object has a hole in its middle, and the segmentation in the image indicates that this hole is also part of the object.

The semantic mapping also has limitations if the localization is not accurate and the covariance matrix does not reflect this inaccuracy. During this wrong situation, the elements mapped could be misplaced.

## 6. Conclusions

We have presented our contributions to 3D segmentation from 2D segmented images acquired with a 3D camera in this work. Our approach does not limit the direct correspondence of image pixels with their 3D coordinates. It solves the problem at the limits of 2D segmentation using a novel spatial expansion algorithm. In the first step of the algorithm, it isolates the part of the point cloud that belongs in the raw to each detection. Selecting a position in space, in the second step, it recursively applies an expansion to close connected points. This process eliminates false positives that may appear in the point cloud.

To validate our approach, we have carried out two experiments. In the first one, we have measured our approach's effectiveness to eliminate false positives, comparing it with

the direct approach and with one that performs a previous erosion in the segmentation in the image. It also showed how our method overperforms in speed compared to other state-of-the-art approaches.

In the second experiment, we have shown how a real environment can be semantically mapped using a mobile robot's location. This method takes into account the uncertainty in estimating the position of a robot. The final map is composed of an octomap where the cells represent the space occupied by the detected object. The color of the cell indicates its class, as well as a probability associated with it.

As future works, we propose to use the semantic map to locate the robot in case of not knowing the initial position of the robot. Another possible extension is to carry out a semantic nevaging, sending the robot positions relative to objects on the stage, such as "near the oven" or "in the kitchen", instead of metric positions.

**Author Contributions:** Conceptualization, F.M.; methodology, F.M. and J.M.G.; software, F.M. and F.G.; validation, J.M.G. and F.G.; formal analysis, M.F.; writing—review and editing, J.G. All authors have read and agreed to the published version of the manuscript.

**Funding:** This work was supported by the EU-funded projects RobMoSys ITP MROS under Grant Agreement No. 732410.

**Institutional Review Board Statement:** Not applicable.

**Informed Consent Statement:** Not applicable.

**Data Availability Statement:** Not applicable.

**Conflicts of Interest:** The authors declare no conflict of interest.

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
