# Peer review of "Semantic 3D Mapping from Deep Image Segmentation"

_applsci, doi:10.3390/app11041953_

Round 1

Reviewer 1 Report

Paper is well written. Please look at these notes:

1. Lines 57-58 "In summary, this paper has two main contributions", are these the aims?

2. Figure 3 is not clear. Explain it better.

3. Has the paper some limitation? Please add a seciond before the Conclusion section.

4. This paper should be add in the contest of neurosciences. Please look and add this paper which is closely related to your paper at lines 253-254:     Perrini P et al. The contribution of Carlo Giacomini (1840-1898): the limbus Giacomini and beyond. Neurosurgery. 2013 Mar;72(3):475-81; discussion 481-2. doi: 10.1227/NEU.0b013e31827fcda3. 

Reviewer 2 Report

This paper presents an approach for Semantic 3D mapping from deep image segmentation. The paper also presents novel spatial expansion algorithm. Further development in this field is necessary for knowledge the point of view detailed linking between 3D mapping and deep image segmentation. 

The article still needs to be modified and explain in a more detailed way. 

Recommendations for addition:
1.    Affiliation does not contain complete information about the authors, please added.
2.    Section „Abstract “:  Extend this part of the paper with specific results of the study. I recommend reworked the abstract to a deeper level with emphasis on the obtained results.
3.    In your study, I miss the section dedicated to methodology, if it is possible, add the workflow of our methodology. Although this paper is very topical, the reader would know the sequence of processing through this overview of methodology.
4.    Figure.3 - Which kind of utility was used to present the point cloud and Boundary points problems I recommend adding this information.
5.    Line 240 - Section “4. Experiments”, I recommend changing the title of this section to Results, as you present the results of your study in this part.
6.    There is a lack of a deeper scientific discussion about the achieved results linked on indexed scientific work in this field. Recommendation, discuss your results with suitable references. This kind of study requires it.
